# HOPFIELD ENCODING NETWORKS

## ABSTRACT

Content-associative memories such as Hopfield networks have been studied as a good mathematical model of the auto-associative features in the CA3 region of the hippocampal memory system. Modern Hopfield networks (MHN) are generalizations of the classical Hopfield networks with revised energy functions and update rules to expand storage to exponential capacity. However, they are not yet practical due to spurious metastable states even while storing a small number of input patterns. Further, they have only been able to demonstrate recall of content by giving partial content in the same stimulus domain and don't adequately explain how cross-stimulus associations can be accomplished, as is evidenced in the Hippocampal formation. In this paper, we revisit Modern Hopfield networks from both these perspectives to offer new insights and extend the MHN model to mitigate these limitations. Specifically, we observe that the spurious states relate to the separability of the input patterns, which can be enhanced by encoding them before storage and decoding them after recall. We introduce a new kind of Modern Hopfield network called the Hopfield Encoding Network (HEN) to enable this and show that such a model can support cross-stimulus associations, particularly between vision and language, to enable recall of memories with associative encoded textual patterns.

## 1 INTRODUCTION

The hippocampal system in the brain is responsible for long-term declarative memory, which involves remembering facts and events (The Human Memory; Wickens & Carswell, 2021). These 'fragments' of individualized memories are believed to be linked together during ingestion of new information for long-term storage and cued recall in the hippocampal circuitry, as proposed in a seminal paper by (Eichenbaum, 2001). It includes the parahippocampal area that consists of the perirhinal cortex that encodes the "what" or identity, the parahippocampal cortex which encodes the "where" or location, and the entorhinal cortex, which acts as the gateway to converge incoming feeds from multiple sensory stimuli (Strien et al., 2009). For any incoming stimulus, the representations forming memories are created within the trisynaptic circuit of the hippocampus consisting of distinct regions, which includes the CA3 region of cells. The existence of recurrent synaptic connections in CA3 led to the hypothesis that CA3 is an auto-associative network similar to the Hopfield network formulation of (Hopfield, 1982). Several key ideas have emerged from the theoretical analysis of Hopfield networks, and these have strongly influenced how neuroscientists analyze memory networks Almeida et al. (2007).

Classical Hopfield networks are dense associate memory architectures (Krotov & Hopfield, 2016) that can store a collection of multidimensional vectors, or memories, as fixed point attractor states of a recurrent dynamical system. Their purpose is to connect the initial or input state to a final state at a fixed point corresponding to a specific memory. Hopfield networks store content implicitly using Hebbian recurrent learning by treating the various patterns as stable basins in an energy landscape and reconstructing them by giving a portion of the content again as a recall cue (Hopfield, 1982). Despite their biological plausibility, Hopfield networks have not seen great adoption in content storage systems due to their limited storage capacity.

The Modern Hopfield Network was introduced as a continuous relaxation of the original Hopfield network from the 1980s and has been shown theoretically to have exponential storage capacity (Krotov & Hopfield, 2016; Demircigil et al., 2017). However, they are not yet practical because they have a predilection to enter spurious metastable states, l eading to memorizing bogus patterns even while

storing a small number of inputs. Further, while there is some evidence of work in cross-stimulus associations in the context of classical Hopfield network (Shriwas et al., 2019), to our knowledge, Modern Hopfield networks have only been able to demonstrate recall of content by giving partial content in the same stimulus domain. Hence, they do not adequately explain how the hippocampal formation can bind across stimulus domains in their revised formulation (Borders et al., 2017).

In this paper, we revisit Modern Hopfield Networks (MHN) to offer new insights and extend the MHN model to mitigate some of its limitations. We first investigate the spurious states problem to put forward the hypothesis that it is related to the separability of the input patterns which can be mitigated by choosing suitable encoded representations of the stimulus for storing in the network. We then advance the hypothesis that such encoded representations of stimuli extend the capabilities of Modern Hopfield Networks to store cross-stimulus associations. We demonstrate this using vision-language associations, choosing both identical and separate encodings of the different stimuli. We thus demonstrate that either strategy works provided the uniqueness of associations is assured (which is still key to content-associative memories). These new Modern Hopfield Network design enhancements could lead to increased adoption of the Hopfield model in practical storage systems.

## 1.1 PRELIMINARIES

We review the basic framework of MHN and encoder-decoder architectures below to motivate the proposed enhancements to MHN using encodings. Further, all our insights are derived from our experiments on the MS-COCO dataset of 110,000 images (Lin et al., 2015). We chose this dataset to base our study as it contained unique associative captions that will be helpful to illustrate the cross-stimuli associations in the Modern Hopfield network.

## 1.2 MODERN HOPFIELD NETWORKS

The mathematical framework of a dense associative memory for a modern continuous Hopfield network can be described in terms of the energy function and the attractor dynamics. Let $N$ be the number of memories and $K$ be the data dimensionality. Defining a similarity metric between the memories $\{\xi_n \in \mathcal{R}^{K \times 1}\}_{n=1}^N$ and the state vector $\mathbf{v} \in \mathcal{R}^{K \times 1}$ the generalized formulation is written as:

$$E = -\log\left(\sum_n \exp(f_{\text{sim}}(\xi_n, \mathbf{v}; \beta))\right) \tag{1}$$

Essentially, the function $f_{\text{sim}}(\xi_n, \mathbf{v}; \beta)$ is a measure of similarity between the state vector and each memory in the bank. The log of the sum of exponential (LSE) metric is designed to select the most relevant memory across the bank (i.e. term with the maximal similarity value), with the parameter $\beta$ controlling the tightness of this approximation. $\beta$ is often referred to as the weighted inverse temperature term and is a hyperparameter for this model.

Modern continuous Hopfield networks (Ramsauer et al., 2020; Krotov & Hopfield, 2020) use a dot product-based similarity, which can be expressed as $f_{\text{sim}}(\xi_n, \mathbf{v}; \beta) = \beta < \xi_n, \mathbf{v} >$. Alternatively, (Saha et al., 2023) use the negative $\ell_2$ distance $f_{\text{sim}}(\xi_n, \mathbf{v}; \beta) = -\beta ||\xi_n - \mathbf{v}||_2^2$ as a measure of similarity. The general form of the state vector updates can be expressed as the following recurrence:

$$\mathbf{v}^{(t+1)} = \mathbf{v}^{(t)} + \left(\frac{\partial t}{\tau_f}\right) \frac{\partial E}{\partial \mathbf{v}^{(t)}} \tag{2}$$

Where $\tau_f$ is the time constant weighting i.e learning rate, which can be set according to (Krotov & Hopfield, 2020). The LSE formulation in Eq. (1) can be expressed as a Lagrangian form in the state of the memory and feature neurons, such that the update dynamics in Eq.( 2) are guaranteed to decrease the energy monotonically (Millidge et al., 2022). Furthermore, theoretical convergence guarantees on the individual update steps have been developed by (Ramsauer et al., 2020).

Given a partial state vector $\mathbf{v}^{(0)}$ as a query (initialization), upon convergence, the Hopfield updates produce a completed pattern $\mathbf{v}^{(T)}$ that should correspond to one of the stored memories $\xi_k$. The number of time steps $T$ is a hyperparameter that is set beforehand.

### 1.2.1 Encoder-Decoder Architectures:

From the perspective of artificial neural networks, a bulk of modern representation learning algorithms rely on encoder-decoder architectures for efficient information processing. A number of encoder-decoder architectures have emerged recently that aim to project the input pattern into a representation space with better separability to allow accurate reconstruction. Essentially, such frameworks learn a mapping back and forth from the input data into a latent space with desirable statistical and geometric properties. A bottleneck layer at the encoder stage forces the representation to be compact and retain salient information that the decoder can then use to faithfully reconstruct the input data. Popular examples of such architectures in active use include Variational Autoencoders (VAEs) (Kingma et al., 2019), Vector Quantized VAEs (Van Den Oord et al., 2017), Generative Adversarial Networks (GANs) (Creswell et al., 2018), VQ GANs (Kumar et al., 2019), and sequence to sequence diffusion models (Croitoru et al., 2023) which have been adopted for a wide-variety tasks such as image denoising and in-painting, text to image generation such as with Discrete VAE (D-VAE) (Ramesh et al., 2021), and natural language processing (Yuan et al., 2022).

## 2 Hopfield Encoding Networks

Although the results of modern Hopfield networks indicate that the network has theoretically exponential capacity to store memories, the system of updates is known to be highly sensitive to the choice of $\beta$, which may vary widely across different data representations and can lead to spurious attractor basins (Bruck & Roychowdhury, 1990; Ramsauer et al., 2020; Barra et al., 2018). As alluded to in (Krotov & Hopfield, 2020), a sufficiently high temperature encourages the energy landscape to follow a peaky distribution around the attractors (memories). On the other hand, a lower value results in a wider spread across the candidate exemplar memories. Regardless of the choice of beta, the Hopfield networks are known to enter spurious attractors states, leading to memorizing bogus patterns due to a combination of similar looking patterns (Ramsauer et al., 2020). Since Hopfield networks rely on evaluated similarity between queries and stored memories for accurate associative memory retrieval, poor direct image similarity assessments tend to produce weak basins of attraction in the Hopfield energy landscape space, leading to meta-stable formulations, particularly for larger image collections. This happens even when storing a small number of patterns with a good choice of $\beta$ ($\beta = 150$) as shown in Fig. 1. Only a small number of memories are recalled accurately, while the rest are resolved to an average image. The spurious attractor states problem has been well-studied, and it was shown as early as in (Hopfield, 1982) that the spurious memories are correlated to the memories being stored in the Hopfield Network, indicating that this may be due to the inherent lack of separability in the input patterns.

The key idea we put forward here is to see if we can increase the separability of the input patterns before they enter the Modern Hopfield network so as to reduce the spurious attractor states problem.

*Hypothesis 1: The spurious attractor states can be reduced by encoding input patterns prior to storing them in the Modern Hopfield network and decoding them after recall.*

Our proposed Hopfield Encoding Network (HEN), therefore, combines an encoder-decoder with the Modern Hopfield network. Instead of storing the raw content, the encodings are generated by a pre-trained auto-encoder and stored in the modern Hopfield network. The raw content can be regenerated through chaining with the decoder portion of the auto-encoder. We hypothesize that encodings produced by a learned auto-encoder for the incoming content contain discriminative information that is not only compact but improves the separability in the energy landscape to prevent metastable states. That is, by leveraging a well-trained auto-encoder for feature extraction, we posit that the most significant and discernible features between images can be easily identified, leading to less spurious patterns emerging during recall.

### 2.1 Choice of encoders

To evaluate this hypothesis, we examine the effectiveness of various pre-trained encoder-decoder architectures to produce encoded representations that can lead to successful recall of dense associative memories. Further, we also analyzed the parameter choices for the energy formulation of dense associative memories in affecting the identity of the recalled memory items when using their encoded representations.

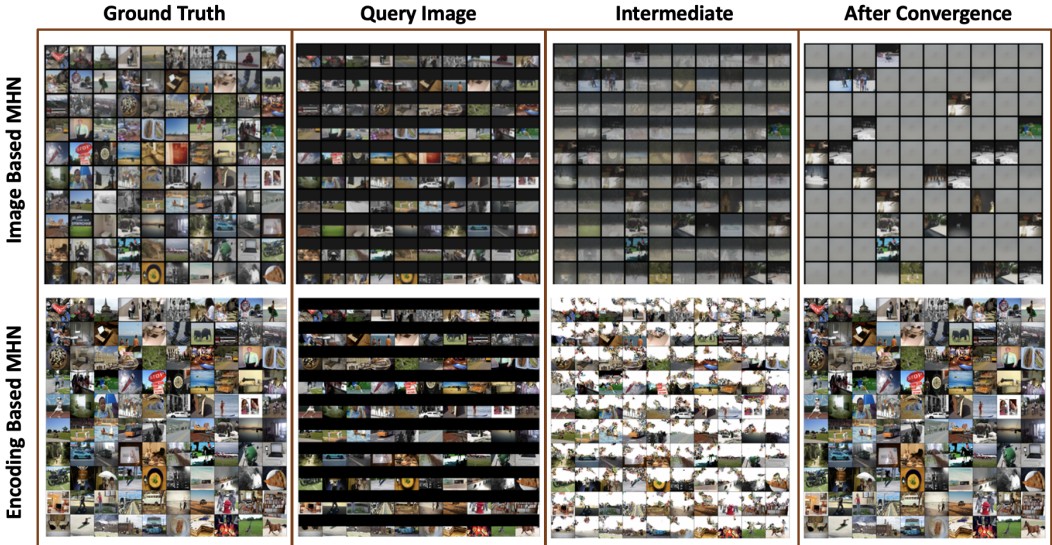

Figure 1: The figure illustrates the progression of the **(Top Row)** Modern Hopfield Network (MHN) run on image inputs and the **(Bottom Row)** Hopfield encoding network at different intermediate steps. The sequence from left to right is as follows: the original image, the query image with half of it occluded, an intermediate update at iteration 11, and the final reconstruction at iteration 150. We use an $\ell_2$ similarity and discrete Variational Autoencoder (D-VAE) encoder for the encoded Hopfield network. For both experiments, we set $\beta = 150$

Specifically, we evaluated various pre-trained encoder-decoder architectures known for their state-of-the-art performance in deep learning-based image encoding and decoding. In particular, we utilized the Discrete Variational Autoencoder (D-VAE) from (Ramesh et al., 2021) and other architectures from (Rombach et al., 2021). We also explored two variants from the diffusion library: one trained with codebook-based criteria and the other using Kullback-Leibler (KL) divergence-based criteria. Our empirical analysis, detailed in a later section, revealed that Vector Quantized VAE (VQ-VAE) methods outperformed others in our setup. Consequently, we selected D-VAE and variants of

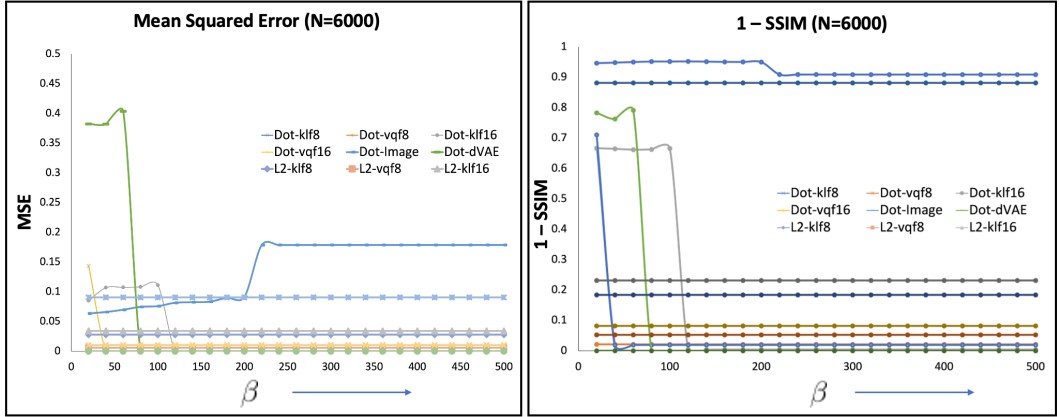

Figure 2: Memory recall performance of various encoder methods and the image-based Modern Hopfield network, each color-coded differently. The **(Left)** figure plots the MSE metric while the **(Right)** depicts the 1-SSIM metric as a function of the temperature $\beta$. The encoder-based methods outperform the raw image-based method over a very large range of choices of hyperparameters. A 6000-image subset of the COCO dataset was used for this experiment.

Table 1: All encoder-based methods can recover image representations without quality loss, even as more images are stored in $\{\xi_n\}$.

| NUM IMAGES 1-SSIM, MSE | vqf8 | vqf16 | Image | D-VAE |
|---|---|---|---|---|
| 6000 | 0.021, 0.000 | 0.019, 0.004 | 0.836, 0.064 | **0.000, 0.000** |
| 8000 | 0.019, 0.000 | 0.046, 0.004 | 0.835, 0.067 | **0.000, 0.000** |
| 10000 | 0.019, 0.000 | 0.047, 0.004 | 0.835, 0.064 | **0.000, 0.000** |
| 15000 | 0.019, 0.000 | 0.048, 0.004 | 0.836, 0.066 | **0.000, 0.000** |

VQ-K8 and VQ-F16 from (Rombach et al., 2021) for further analysis. To maintain consistency in representation, we downsampled the images to a resolution of $28 \times 28 \times 3$.

### 2.1.1 REDUCTION OF METASTABLE STATES DUE TO USE OF ENCODINGS

This study tested the image-based dense associative memories against pre-trained Discrete VAE (Ramesh et al., 2021) and VQ-VAEs encoding equipped Hopfield encoding network (Rombach et al., 2021). The test was conducted on a memory bank ($\{\xi_n \in \mathcal{R}^{1 \times K}\}$) and query size ($N$) of 6000 images from the MS-COCO dataset. Both dot product and $\ell_2$ based similarity measures were attempted. The performance of different encoder-decoder architectures was evaluated (see Fig. 2). by varying the dimensionality $K$. The Mean Squared Error ($MSE$) and Structural Similarity Index ($1 - SSIM$) metrics were used to compute the similarities between the encoder reconstructions stored in the $\{\xi_n\}$ memory bank and the reconstructed ones.

Note that as this evaluation was conducted to assess the performance on metastable states, *the focus was on recovering the correct identity rather than the quality of reconstruction.* Hence a $MSE = 1 - SSIM = 0$ indicated that the dense associative memory could retrieve the full encoded representation of the image from which the pre-trained decoder could reconstruct the image. The results in Figure 2 show that using the encoded representations of input patterns to store in the Hopfield network results in fewer metastable states. Further, the more separable the embedding, the larger the dataset over which Hopfield networks can avoid getting into metastable states.

Fig. 1 shows the result of using D-VAE encoding for perfect memory recall for the same set of images for which raw image storage in the Hopfield network failed. While the quality of the reconstruction is not as clear as the original, the identity of the images recovered is preserved one-to-one. In comparison, the recall using the raw images for the same dataset using the Modern Hopfield network shows the metastable states. Further, we studied the performance of the HEN on an increasing number of images in the memory bank. Table 1 shows the performance metrics across different encoder approaches with an increasing number of images. All encoder-based approaches robustly recover the encoder image representations without seeing a drop in the quality of the reconstruction with an increasing number of images stored in $\{\xi_n\}$.

### 2.1.2 SEPARABILITY ACHIEVED IN ENCODINGS

To study the extent of separability achieved in various encodings, we examine the strength of association patterns in the memory bank of the Hopfield encoding network. Extending the notation in Section 1.2, let $\mathbf{V}_i^{(0)} \in \mathcal{R}^{1 \times K}$ denote the encoded query for example $i$ in the dataset. This encoding is generated by occluding a portion of the image fed through the encoder (or just the occluded image for the raw image MHN). Specifically, we expect that the major contributor to poor recovery performance is the lack of separation between the attractor basins in the energy landscape in Eq. (1), due to which the dynamics governing the evolution of $\mathbf{V}_i^{(0)}$ in Eq. (2) default to meta-stable configurations. This is also alluded to in (Hopfield, 1982).

To quantify this separation, we compute the cosine similarity between pairs of query and memory vectors, i.e. $c_{ij} = \cos(\mathbf{V}_i^{(0)}, \xi_j) = \mathbf{V}_i^{(0)} \xi_j^T / ||\mathbf{V}_i^{(0)}||_2 ||\xi_j||_2$. If the patterns of association are well separated, each query $\mathbf{V}_i$ as projected in the encoding space (or in the native space for raw images) is close to its own memory $\xi_i$ but is far apart from other memories $\xi_j, \forall j \neq i$. We test this hypothesis

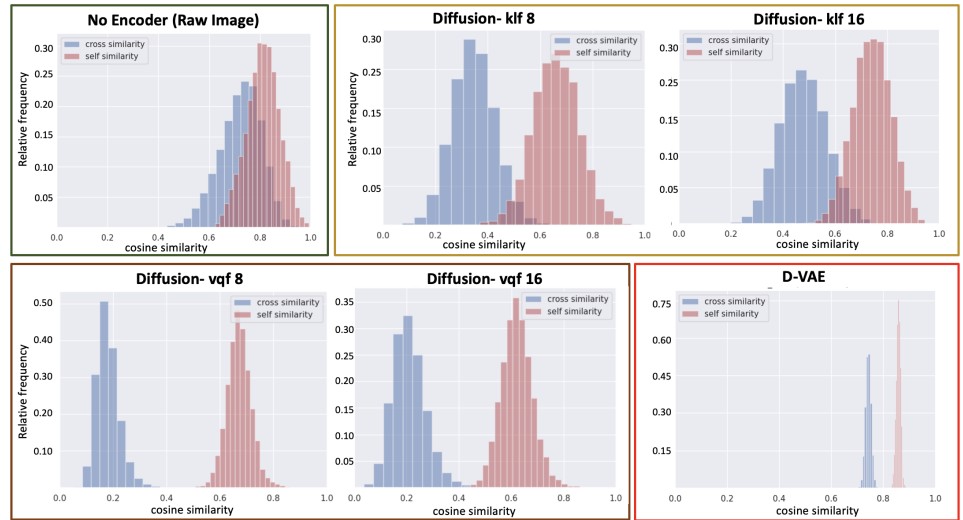

Figure 3: Illustrating separability in various embeddings used for the experiment in Subsection 2.1.1. In each figure, we plot the histogram of the distribution of cosine similarity values between the queries and memories, i.e. $\cos(\mathbf{V}_i^{(0)}, \xi_j)$. Distributions colored in red indicate self-similarity (i.e. $i = j$) across paired examples, while distributions colored in blue indicate cross similarities (i.e. $i \neq j$). We generate these distributions for (a) Raw Images in the Black Box, (b) Diffusion models trained on KL Divergence in the Orange Box, (c) Diffusion Models trained using Vector Quantization in the Brown Box, and (d) Discrete-VAE (D-VAE) in the Red Box. We observe that the separability in cross and self-similarity measures is particularly poor in the case of raw images.

in Fig. 3 by plotting the distribution of values $c_{ij} \forall j \neq i$ as a histogram colored in blue and $c_{ii}$ as a histogram colored in red, for all choices of encoders $\mathbf{\Phi}_{\text{enc}}(\cdot)$ in the Hopfield Encoding Network. As can be seen in Fig. 3, this separation is poor for the raw image case, with the two histograms having a high overlap in values. This overlap substantially reduces across the encoder-based models, with the Vector Quantized variants providing improved separability and a relatively higher magnitude of self-similarity values compared to their KL counterparts. Finally, we notice that the D-VAE encoder, besides providing separable encodings, also results in the tightest fit around the mean for the self and cross-similarity value distributions. This is also the reason why the D-VAE encoder gave the best performance in avoiding metastable states as shown in Fig. 2 and Table 1.

## 3 HOPFIELD ENCODING NETWORK SUPPORTS CROSS-STIMULI ASSOCIATIONS

So far, Hopfield networks have been used to recover given stimuli using partial cues coming from the same stimuli type, even if the ultimate form in which they are stored is in encoded representations. As the hippocampal system can do cross-associations between stimuli, we explore next if the Hopfield encoding network (HEN) framework can also support this type of recall. We note here that the cross-associative features have been previously demonstrated for the classical Hopfield networks model, which required the binarization of patterns (Shriwas et al., 2019), limiting both the scalability and reliability at retrieval as well as a direct application to the continuous representations such as encodings. Specifically, we explore the use of cross-stimuli coming from language and vision, as language-based queries are easier to use as cues for recall in practical storage systems.

*Hypothesis 2: Hopfield encoding networks serve as content-addressable memories even with cross-stimuli associations as long as they are unique associations.*

To validate this hypothesis, we conducted three separate experiments. First, we stay within the paradigm of similar stimuli type as a cue and render the language cue into the familiar image form to allow for content-based access. In the second, we use a native textual embedding for the language cue and associate it with the content to be stored. Finally, we show that if the uniqueness of asso-

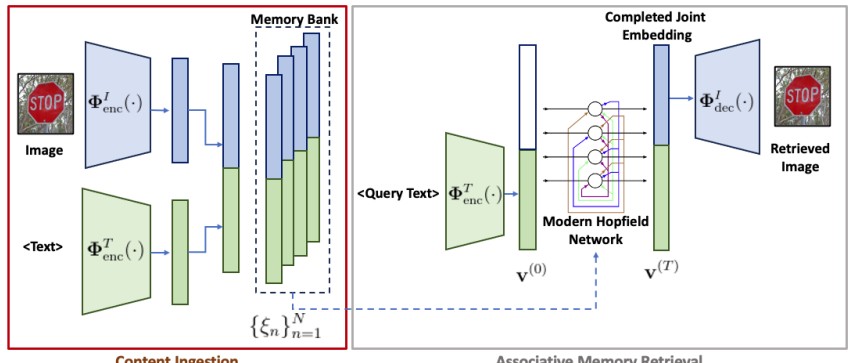

Figure 4: Hopfield Encoded Network (HEN) architecture. **Orange Box:** At ingestion, paired text and image inputs are fed to text ($\mathbf{\Phi}_{\text{enc}}^{T}(\cdot)$) and image ($\mathbf{\Phi}_{\text{enc}}^{I}(\cdot)$) encoders respectively to generate the memories of the HEN memory bank. **Grey Box:** At query time, a partial query $\mathbf{v}^{(0)}$ is generated by feeding the query text into $\mathbf{\Phi}_{\text{enc}}^{T}(\cdot)$. After ($T$) iterations of Eq. (2), the image encoding is extracted from $\mathbf{v}^{(T)}$, and decoded through the Image decoder ($\mathbf{\Phi}_{\text{dec}}^{I}(\cdot)$) to retrieve the corresponding image. Using full image representations instead of image encodings is equivalent to setting $\mathbf{\Phi}_{\text{enc}}^{I}(\cdot) = \mathbf{\Phi}_{\text{dec}}^{I}(\cdot) = \mathcal{I}$, the identity transformation. In the experiment where the text captions are pixelized as input, $\mathbf{\Phi}_{\text{enc}}^{T}(\cdot) = \mathbf{\Phi}_{\text{enc}}^{I}(\cdot)$

ciation is lost, spurious memory states could again emerge even if the inputs are encoded. Fig. 4 illustrates the overall methodology for the Hopfield Encoding Network (HEN). Here, the memory bank vectors are formed by concatenating image and text embeddings $\xi_n = [\mathbf{\Phi}_{\text{enc}}^{\mathbf{I}}(\mathbf{I}_n); \mathbf{\Phi}_{\text{enc}}^{\mathbf{T}}(\mathbf{T}_n)]$. During retrieval, we construct a query vector $\mathbf{v}^{(0)} = [\mathbf{0}; \mathbf{v}_T]$ constructed using the encoded text vector $\mathbf{v}_T = \mathbf{\Phi}_{\text{enc}}^{\mathbf{T}}(\mathbf{T}_n)$ and zeros in the location of the image encodings. Finally, after convergence $\mathbf{v}^{(T)} = [\mathbf{v}_I^{(T)}; \mathbf{v}_T^{(T)}]$, we decode the image embedding $\mathbf{\Phi}_{\text{dec}}^{\mathbf{I}}(\mathbf{v}_I^{(T)})$ to retrieve the image content.

## A. HANDLING CROSS-STIMULI USING PIXELIZED LANGUAGE-IMAGE ASSOCIATION

To test language-image associations, we utilized the unique set of captions associated with each image in the COCO dataset. Specifically, we employed Python's `hashlib.sha256(.)` function to hash the captions generating a unique ID text string to associate with the image. Initially, we created a memory bank $\{\xi_n\}$ by converting the hashed captions into pixelized text representations using a generic text-to-pixel function. Subsequently, both the pixelized text and the corresponding images were processed through the same encoder. The resulting vectors were concatenated to form the elements of the memory bank $\{\xi_n\}$.

During the query phase, we supplied only the pixelized text part of the encoded vector, setting the image component to zero. The Hopfield network iteratively updated the image encoding vector, which was passed through the corresponding decoder to reconstruct the image. Fig. 5 illustrates the network's progression in reconstructing the image based on the pixelized text input. The recurrent updates in the Hopfield network iteratively reconstructed the full image.

As illustrated in Figs. 5, the HEN network is able to recall perfectly using pixelized cross-stimuli associations. Table 2 reveals that all HEN variants with different encodings still outperformed traditional image-based Hopfield networks even as the number of image patterns to store increased.

## B. NECESSITY OF UNIQUE TEXT-IMAGE ASSOCIATIONS FOR ACCURATE RECOVERY

While the HEN can recall accurately based on cross-stimuli associations, we expect such associations to be unique as in the case of stimuli from the same domain/modality.

*Hypothesis 3: Cross-stimuli associations must be unique in order to avoid metastable states during recall.*

To validate this hypothesis, we designed an experiment in which two different images to be stored in HEN were selected at random and associated with the same textual pattern. Subsequently, we

Table 2: The table displays the performance of various encoded cross-modal Hopfield encoding networks compared to image-based Modern Hopfield networks as the memory bank $\{\xi_n\}$ increases. All of the different encoders performed well. The first line shows the CLIP-encoded cross-modal representations, while the following lines present pixelized text-encoded representations for increasing image sizes.

| NUM IMAGES 1-SSIM, MSE | vqf8 | vqf16 | Image | D-VAE |
|---|---|---|---|---|
| 6000-CLIP | 0.016, 0.000 | 0.024, 0.000 | 0.681, 0.118 | **0.000, 0.000** |
| 6000 | 0.016, 0.000 | 0.024, 0.000 | 0.952, 0.214 | **0.000, 0.000** |
| 8000 | 0.016, 0.000 | 0.023, 0.000 | 0.952, 0.215 | **0.000, 0.000** |
| 10000 | 0.015, 0.000 | 0.024, 0.000 | 0.952, 0.215 | **0.000, 0.000** |
| 15000 | 0.015, 0.000 | 0.024, 0.000 | 0.952, 0.215 | **0.000, 0.000** |

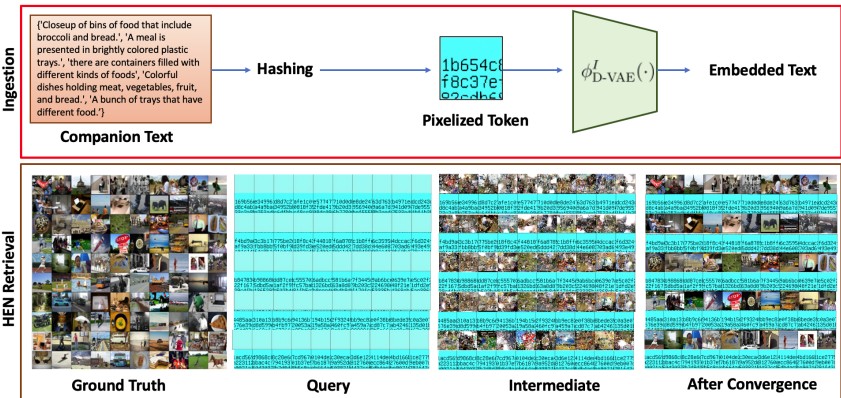

Figure 5: Step-by-step progression of a cross-modal query using only text input. **Top Row:** The companion text is pixelized and encoded (using D-VAE in this example). This encoded representation is used to reconstruct the complete image. **Bottom Row:** The reconstruction process for the experiment in Subsection 3. **(L-R)** Ground Truth, Iteration $t = 0$ starting with a blank canvas with the provided pixelized text inputs as query prompts, an intermediate update, and finally the fully reconstructed image. This visualization effectively demonstrates the network's ability to accurately reconstruct the image from a text-only input modified into an image-based representation

queried the system using the pixelized text to observe the type of images that would be reconstructed. Our findings shown in Fig. 6 validate our hypothesis. Violating the uniqueness constrain led to spurious recall where the reconstructed image appeared to be a mixture of two different images. Thus HEN supports cross-stimuli associations and the recall is accurate if the associative text pattern is distinct per image to be stored.

## C. Handling cross-stimuli using separately encoded language-image associations

Next, we explored if it was necessary to have the cross-stimulus be rendered in the same form as the memory patterns for HEN. Specifically, we conducted an experiment in which different encodings were used to represent images and text. We retained the best performing encoder (D-VAE) for image encoding (See Table. 5) but the textual associative stimulus was encoded using the CLIP foundational model (Radford et al., 2021). Our hypothesis posited that if the norm spaces of these encodings are similar, the unique association should still hold.

To create a more meaningful embedding, we concatenated the set of caption sentences per image into a single long sentence. This sentence was then encoded using the pre-trained CLIP model. The resulting text and image encodings were then ingested into HEN, as illustrated in Fig.7 (Top). The

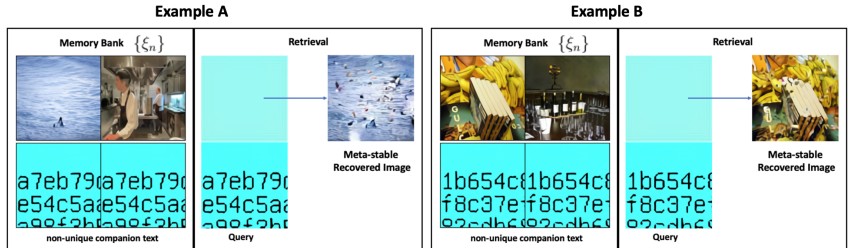

Figure 6: The figure illustrates the consequences of breaking unique text-image associations via two separate examples. In each example, **(Left)** the upper and lower image cases depict the disruption of unique associations in the $\{\xi_n\}$ memory bank. **(Right)** A single text input corresponding to these disrupted associations is used during the query phase. The top blue image represents an empty image as a zero-encoded vector at initialization. The resulting reconstruction appears to be a meta-stable state. Neither of the original images is accurately recovered, validating our hypothesis on the importance of unique text-image associations.

experiment yielded promising results. The performance of the 6,000 images tested was on par with that of the discrete VAE in the same embedding space. This is a significant finding, suggesting that text and image encoders can operate in disjoint spaces while still achieving accurate reconstructions, provided the Hopfield energy landscapes can be normalized. Further, the top row of Table. 2 shows robust performance across all CLIP combinations and other image-encoded representations.

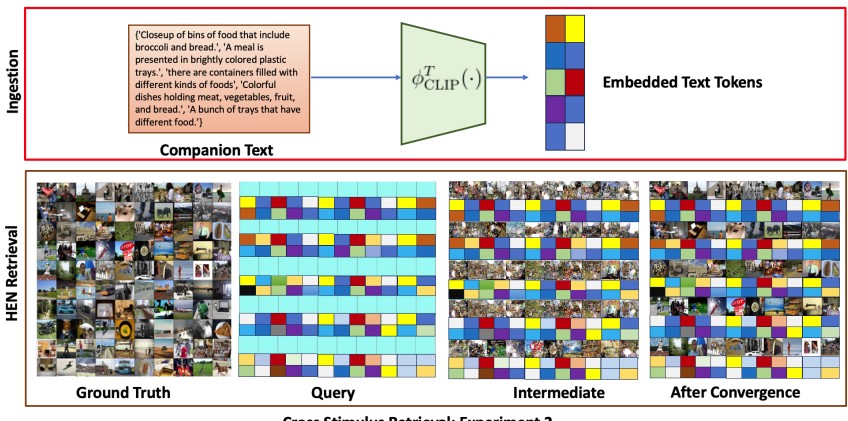

Figure 7: Step-by-step progression of a cross-modal query using two different encodings for associated visual and language cues. **Top Row:** The text stimulus is encoded via the CLIP (Radford et al., 2021) text encoder and associated with the image represented by a D-VAE encoded vector. **Bottom Row:** The reconstruction process for the experiment in Subsection 3. **(L-R)** Ground Truth, Iteration $t = 0$ starting with a blank canvas with the provided CLIP Encoded text inputs as query prompts, an intermediate update, and finally the fully reconstructed image. This visualization effectively demonstrates the network's ability to accurately reconstruct the image from a text-only input from a completely different stimulus space as the image content.

## 4 CONCLUSIONS

In this paper, we extend the Modern Hopfield Network formulation with two key enhancements, namely, introducing pattern encoders and decoders in combination with the Modern Hopfield network to improve the separability of patterns and the reduction of metastable states. We also show how such a network is capable of cross-stimuli associations using differing encodings for different stimuli as long as the uniqueness of association criteria is met. These new enhancements can make Modern Hopfield networks one day practical for content storage systems.

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
