# OpenReview forum: "Hopfield Encoding Networks"
_ICLR.cc/2024/Conference — Submitted to ICLR 2024_

### Official Review · Reviewer_hRE9 · 2023-10-30

**Soundness:** 1 poor
**Presentation:** 1 poor
**Contribution:** 2 fair
**Rating:** 3
**Confidence:** 4

**Summary:**

At a high level, the intuition for this paper is logical: by increasing the differences between stored patterns in an associative memory network (by projecting them into a more separating/distinguishing space, in this case using autoencoders), the network can more easily recall the patterns.

**Strengths:**

The intuition and basic idea is good. There is testing on real-world data. The text and figures are mostly comprehensible.

**Weaknesses:**

**Summary**

The testing of this idea is very limited and not rigorous. Normally, for a paper in this field, there is at least more systematic testing, diverse datasets, and/or theoretical contributions (i.e., proofs). But this paper lacks those, and only addresses the topic superficially. Add to this the many mistakes and misleading claims, and none of the stated conclusions (bottom of page 9) are well-supported or justified. Accordingly, I cannot rate this submission too highly.

**Ignored hetero-association literature**

From the abstract: “[modern Hopfield networks] have only been able to demonstrate recall of content by giving partial content in the same stimulus domain and don’t adequately explain how cross-stimulus associations can be accomplished, as is evidenced in the Hippocampal formation”

From the top of page 2: “while there is some evidence of work in cross-stimulus associations in the context of classical Hopfield network (Shriwas et al., 2019), to our knowledge, Modern Hopfield networks have only been able to demonstrate recall of content by giving partial content in the same stimulus domain. Hence, they do not adequately explain how the hippocampal formation can bind across stimulus domains in their revised formulation (Borders et al., 2017)”

From the first paragraph of section 3, on page 6: “We note here that the cross-associative features have been previously demonstrated for the classical Hopfield networks model, which required the binarization of patterns (Shriwas et al., 2019), limiting both the scalability and reliability at retrieval as well as a direct application to the continuous representations such as encodings”

The central claim here (that hetero-association work, here called “cross-stimulus association”, is very limited and restricted to binary data) is quite misguided and wrong, both from machine learning and neuroscience perspectives. Dense associative memory networks (modern Hopfield networks) have been used to study hetero-association, and there is a very long history of more classical models doing the same. Additionally, since the claim and paper are motivated by hetero-association in the hippocampal formation (and this point is emphasised at multiple points), it seems very odd not to mention the related work done by neuroscientists. Below are references which illustrate literature (or content therein) this submission ignores. I encourage the authors to read these papers and re-evaluate their claims, making a careful attempt at finding what is different and new in their own work. Relatedly, “hypothesis 2” and “hypothesis 3” needs to be re-presented and put into their proper existing contexts.

*Classical hetero-associative work (non-exhaustive list)*

S.-I. Amari. Learning patterns and pattern sequences by self-organizing nets of threshold elements. IEEE Transactions on Computers, C-21(11):1197–1206, 1972. doi: 10.1109/T-C.1972.223477

H. Gutfreund and M. Mezard. Processing of temporal sequences in neural networks. Phys. Rev. Lett., 61:235–238, Jul 1988. doi: 10.1103/PhysRevLett.61.235

M. Griniasty, M. V. Tsodyks, and Daniel J. Amit. Conversion of Temporal Correlations Between Stimuli to Spatial Correlations Between Attractors. Neural Computation, 5(1):1–17, 01 1993. ISSN 0899-7667. doi: 10.1162/neco.1993.5.1.1

*Modern work with dense networks*

Danil Tyulmankov, Ching Fang, Annapurna Vadaparty, and Guangyu Robert Yang. Biological learning in key-value memory networks. In M. Ranzato, A. Beygelzimer, Y. Dauphin, P.S. Liang, and J. Wortman Vaughan (eds.), Advances in Neural Information Processing Systems, volume 34, pp. 22247–22258. Curran Associates, Inc., 2021.

Millidge et al. ICML 2022 (this paper is already cited in the submission, but without reference to its hetero-association result, see its appendix)

Arjun Karuvally, Terrence Sejnowski, and Hava T Siegelmann. General sequential episodic memorymodel. In Andreas Krause, Emma Brunskill, Kyunghyun Cho, Barbara Engelhardt, Sivan Sabato, and Jonathan Scarlett (eds.), Proceedings of the 40th International Conference on Machine Learning, volume 202 of Proceedings of Machine Learning Research, pp. 15900–15910. PMLR, 23–29 Jul 2023.
Hamza Chaudhry, Jacob Zavatone-Veth, Dmitry Krotov, Cengiz Pehlevan, Long Sequence Hopfield Memory, NeurIPS 2023.

*Related neuroscience work*

Whittington, James CR, et al. "The Tolman-Eichenbaum machine: unifying space and relational memory through generalization in the hippocampal formation." Cell 183.5 (2020): 1249-1263.

Maxwell Gillett, Ulises Pereira, and Nicolas Brunel. Characteristics of sequential activity in networks with temporally asymmetric hebbian learning. Proceedings of the National Academy of Sciences, 117(47):29948–29958, 2020. doi: 10.1073/pnas.1918674117.

Thomas F. Burns, et al. Multiscale and Extended Retrieval of Associative Memory Structures in a Cortical Model of Local-Global Inhibition Balance, eNeuro 23 May 2022, 9 (3) ENEURO.0023-22.2022.

**Similarity metrics**

In the third paragraph of section 1.2, there is the sentence “Alternatively, (Saha et al., 2023) use the negative $ℓ_2$ distance $f_{sim}(ξ_n, v; β) = −β||ξ_n − v||_2^2$ as a measure of similarity.”

However, a far larger collection of similarity metrics were tested by Millidge et al., 2022. Higher dimensional metrics were also tested by Burns & Fukai ICLR 2023. Subsequently, it seems odd to select only Saha et al. 2023 to mention. I suggest removing the sentence or also discussing the wider range of other similarity metrics. Probably the best option is to actually tell us why you choose the distances you do.

**Ignored other encoding methods**

The authors should also survey and discuss how their encoding method is different to others, e.g.,
Louis Kang, Taro Toyoizumi, A Hopfield-like model with complementary encodings of memories, arXiv:2302.04481

**Incorrect and imprecise statements**

From paragraph 2, page 1: “Classical Hopfield networks are dense associate memory architectures”

Most authors (including Hopfield himself) do not equate the “classical Hopfield network” (which has an energy function of $E=-\sum_{\mu=1}^P (\xi^{\mu} S)^2$ and the “dense associate memory architecture” (which has an energy function of $E=-\sum_{\mu=1}^P F(\xi^{\mu} S)$. While it is true that when $F(x)=x^2$, the dense associative memory network becomes the classical one popularized by Hopfield’s work, the sentence above is misleading and wrong without this context.

The description of convergence in associative memory networks in the final paragraph of page 2 implies that any “choice” of $T$, any update rule/energy function, and any set of patterns provides good convergence to individual patterns. This is fundamentally incorrect – how long and whether the dynamics leads to fixed points, whether those will exactly be the stored patterns, and so on, is totally dependent on the originating energy function, chosen patterns, and the way in which the testing is done, e.g., how much the patterns are perturbed.

**“Problem” of meta-stable states**

From the abstract: “they are not yet practical due to spurious metastable states even while storing a small number of input pattern”
From paragraph 3, pages 1-2: “they have a predilection to enter spurious metastable states, leading to memorizing bogus patterns even while storing a small number of inputs”

What and where is the evidence of this? The first paragraph of section 2 (page 3) seems to argue that the evidence for this is because some choices of $\beta$ drive the dynamics towards more meta-stable states which mix individual patterns. However, this doesn’t significantly limit practical use: a sufficiently low value of $\beta$ will work for a given dataset and model. Suggesting this is a common problem (or a problem at all) is therefore unsupported for dense networks. It is true that this was a big problem for classical networks, and was widely studied. Figure 1 is offered as some empirical evidence of failure of dense networks given a particular choice of $\beta$, but this is just one example. Empirical/numerical evidence which would support the claim of this being a problem would need to at least systematically vary $\beta$, the total memory load, and different datasets to demonstrate that there is rarely (if ever) a good choice of $\beta$. Showing then that the proposed method in this submission can achieve a reliably higher performance could then be reviewed. Figure 1 is insufficient.

Section 2.1.1 claims that the results demonstrate a “reduction in the number of metastable states”. No, they do not, they demonstrate differences in recall performance between the proposed methods’ variations (and variants in Figure 2) and a particular example of a dense network (in Figure 1). There is no attempted quantification of the number of metastable states, nor is there a systematic difference demonstrated between the proposed method and dense networks.

**Questions:**

In what sense is $T$ generally a “choice” or a “pre-set hyperparameter”?

in section 2.1.2, why choose cosine similarity? How do your results depend on this choice?

---

### Official Review · Reviewer_Xb9Q · 2023-10-31

**Soundness:** 2 fair
**Presentation:** 2 fair
**Contribution:** 1 poor
**Rating:** 3
**Confidence:** 3

**Summary:**

The present study feeds the latent representation of encoders into a continuous Hopfield network model and claims that it improves the recall success of the network model. The present study further uses the cross-modality associations to demonstrate the model

**Strengths:**

It seems novel that the cross-modality association by feeding the latent representations of two encoders from different sensory modalities and into a continuous Hopfield network. The study also compares the performance of the Hopfield network with different encoders, and numerically verifies the recall performance of the Hopfield network depending on the separability of inputs (latent representation of encoders).

**Weaknesses:**

### Major
I feel the biggest weakness is that the improvement of the model is not from modifications of Hopfield model itself, although the paper is entitled "Hopfield encoding network", but instead comes from encoders that provides inputs to the Hopfield network. The continuous Hopefield model is exactly the same as before. Moreover, the paper fails to provide deep insight into how to improve the separability of inputs received by the Hopfield network.

Moreover, considering the similarity of the continuous Hopfield network with other energy-based models such as Boltzmann machine, I will be surprised if no earlier studies fed encoders' outputs into energy-based models. The author didn't compare the present model with other models so I am not confident about the novelty.

### Writing
- I suggest the author change the name of "cross-stimulus associations" to "cross-modality associations".
- Fig 3 doesn't have figure indices.
- The bottom line on page 1: should "l reading" be "leading"?

**Questions:**

NaN

---

### Official Review · Reviewer_TATU · 2023-11-01

**Soundness:** 2 fair
**Presentation:** 3 good
**Contribution:** 1 poor
**Rating:** 3
**Confidence:** 3

**Summary:**

This paper extends Modern Hopfield Networks (MHN) by an encoder and decoder model. The encoder and decoder models are obtained e.g. from auto-encoders such as VAEs. The encoder serves to provide encodings of patterns that hopefully lead to better separability in the associative space of the MHN than the original patterns would. This leads to less spurious correlation and better separated retrieved patterns. The decoder can subsequently be used to decode the retrieved patterns.

**Strengths:**

### Quality
1. Visual comparison of retrieved images and target images are provided via images, cosine similarity histograms, and loss.
2. Cross-stimuli associations are considered.
3. A wide range of temperature values was considered.

### Clarity
4. The proposed idea in the paper is easy to follow. The paper is well written.

**Weaknesses:**

### Originality
5. I have serious concerns regarding the originality of this work, the way the paper is currently worded. I would strongly suggest to address the following questions in the paper:
    1. The transformer attention mechanism has been shown to correspond to continuous MHNs [1]. Transformer architectures [2] contain multiple hidden layers, before and after the attention modules. These hidden layers can be pre-trained or trained end-to-end. I don’t see novelty in the proposed “encoder -> MHN -> decoder” method because “hidden layer → attention → hidden layer” is not novel. [1, 3] explicitly use multiple hidden layers as mapping to the associative space, which seems to be the purpose of the encoder in the proposed method. Furthermore, [4] use an auto-encoder based approach. How is the proposed method different to these architectures, except for the smaller architecture?
    2. Transformer-based models like BERT or ChatGPT are widely used for generative tasks. They are often trained or pre-trained using masking-out and next-token-prediction, which are retrieval/reconstruction tasks. I.e. this is a content-storage-system that is already practical. What is the difference to the application of the proposed method?
    3. Cross-stimuli associations and even cross-domain associations were also considered in [5]. What additional contributions does this paper provide?

### Quality
6. “Hypothesis 1: The spurious attractor states can be reduced by encoding input patterns prior to storing them in the Modern Hopfield network and decoding them after recall.” - I think this hypothesis needs to be reworded, I am not sure what the authors intend to contribute here.
It is known and clear that patterns that are encoded in a way that separates them better in the associative space lead to better retrieval. In the literature referenced below, these encodings (aka mappings) are learned end-to-end to reduce the prediction or reconstruction loss. However, whether the spurious attractor states can be reduced strongly depends on the encoding itself. As extreme example, one could use random encodings and corresponding decodings, which would, in my understanding, not reduce the spurious attractor states. Is the hypothesis here that this always holds for pre-trained (on a similar domain) auto-encoders?
7. I would suggest to also consider the following in the experimental setup:
    1. More datasets, if possible not only image data.
    2. Out-of-domain usage of the auto-encoders. E.g. a type of image data that the encoder/decoder were not trained on. This is to ensure that the statements about reduction of spurious attractor states hold not only for domain-specific auto-encoders. Alternatively, end-to-end fine-tuning or training of the auto-encoder on the domain data would be an option, to ensure the auto-encoder is always domain-specific.

### Clarity
8. I would suggest to clarify if/that train-test leakage does not occur in the paper. I.e. were the auto-encoders pretrained on data that was used to evaluate the models in the figures and tables?

### Significance
9. I am unsure about the significance of this work, as the novelty of the contribution is not clear to me in its current form.

### References
[1] Ramsauer, H., Schäfl, B., Lehner, J., Seidl, P., Widrich, M., Adler, T., ... & Hochreiter, S. (2020). Hopfield networks is all you need. arXiv preprint arXiv:2008.02217.
[2] Vaswani, A., Shazeer, N., Parmar, N., Uszkoreit, J., Jones, L., Gomez, A. N., ... & Polosukhin, I. (2017). Attention is all you need. Advances in neural information processing systems, 30.
[3] Widrich, M., Schäfl, B., Pavlović, M., Ramsauer, H., Gruber, L., Holzleitner, M., ... & Klambauer, G. (2020). Modern hopfield networks and attention for immune repertoire classification. Advances in Neural Information Processing Systems, 33, 18832-18845.
[4] Wang, K., Reimers, N., & Gurevych, I. (2021). Tsdae: Using transformer-based sequential denoising auto-encoder for unsupervised sentence embedding learning. arXiv preprint arXiv:2104.06979.
[5] Jaegle, A., Gimeno, F., Brock, A., Vinyals, O., Zisserman, A., & Carreira, J. (2021, July). Perceiver: General perception with iterative attention. In International conference on machine learning (pp. 4651-4664). PMLR.

**Questions:**

My main question about this work would be a clearer placement in the current field of research and consequently a clearer outline of the novel contributions (see point 5.).

---

### Author Response · Authors · 2023-11-23
**Reviewer Response**

We want to thank the reviewers for their detailed feedback. We appreciate the effort they put into giving us clear and constructive comments. The main point of our work was to enable modern Hopfield networks to be flexible and take pre-trained encoded representations and show that such a model can support cross-stimulus associations, particularly between vision and language, to enable recall of memories with associative encoded textual patterns.

We appreciate the comments provided by the reviewers regarding the need for more precise descriptions of our proposed contributions and the language used to describe them in our paper. They also suggested conducting more experiments to substantiate our claims and including additional datasets and domains to validate our methods. We also acknowledge the importance of further examining the literature on hetero-associative memories. As a result, we have decided to withdraw our submission. We want to thank the reviewers once again for their valuable feedback. However, since ICLR allows public submissions, we would like to clarify some of the reviewer's comments for posterity.

**Clarification points to the comments made by the reviewers:**

- There have been concerns raised by R1 about the relevance of our proposed work to attention-based mechanisms. In Ramsauer et al.'s work, the Hopfield model is used as an intermediate store layer that takes a single update Hopfield layer step to enhance the transformer's long-range attention mechanism. However, our use case differs since we aim to use the Hopfield Network as a memory bank to create unique associations between different encodings across various modalities. We want to investigate how these encodings can retrieve information across modalities and present an interesting study comparing encoded representations. While there may be similarities, our study differs from the work of Ramsauer et al.. Our primary goal is to achieve accuracy in cross-modal/hetra-associative reconstruction. We demonstrate that the encoded representations in the Hopfield network can retrieve information across modalities more effectively and are flexible enough to handle different encodings.

- We respectfully disagree with R1's comparison of our work to the ChatGPT/BERT style of work. Our work falls into the storage and retrieval space, where you return the exact content stored in memory. On the other hand, ChatGPT is a generative prediction model that generates the next token based on extensive training, compute resources, and a large amount of data. However, there are several well-known issues with generative models, such as hallucinations, inability to produce the exact content, producing wrong content, and biased responses. Our work showed we could retrieve the exact content when given a cross-modal partial embedding. Our central reasoning was that you would have the whole content if you had partial information about one modality. We demonstrated that cross-modal works.

- In response to R3's feedback, our paper intended to refrain from claiming that any choice of T, update rule/energy function, and set of patterns will provide good convergence to individual patterns using associative memory networks. Instead, our point was to highlight that the energy formulations for such networks are highly dependent on the type of embeddings, the separability of the dynamics, and the patterns used. Based on our observations, we found that beyond a certain number of iterations T(in our case, 100), the dynamics of the updates did not change and either converged to meta-stable states or achieved perfect reconstruction. We agree with the reviewer's point that the effectiveness of T varies depending on the situation, and it is generally a hyper-parameter that needs to be set to determine the number of iteration steps required for optimal performance.

- Regarding Millidge et al.'s work, R3 commented on not citing their hetra-association experiments. They provided one half of an image and retrieved the other half, but our definition of cross-modality involves differences in modality. Therefore, we conducted extensive experiments associating text and images across different modalities. This is why we did not cite their experiment. However, we plan to correct this by comparing similar experiments.

---

> ### Comment · Reviewer_hRE9 · 2023-11-26
> **Thanks and clarification**
>
> I wish to thank the authors for providing a constructive reply in response to the reviewers' comments, in spite of the non-positive feedback and their decision to withdraw the paper. I highly encourage the authors to continue on this line of work, as I think their high-level intuition is logical and could potentially help lead to interesting discoveries for scientists and more performant models for engineers.
>
> It may be a misunderstanding on my part, but I wish to further clarify one point the authors mention regarding "choices of T" and "treating T as a hyper-parameter": From the neuroscience and statistical physics perspective, I do not think many would consider T a "choice" or a "hyper-parameter", but rather just property of the dynamics and in particular the convergence time. However, I can see from a machine learning perspectives how this might be treated as a chosen hyper-parameter. In future submissions, I suggest the authors carefully consider and explain these different perspectives.

---

### Meta-Review · Area_Chair_kVWT · 2023-12-05

**Metareview:**

All the 3 referees confidently recommended rejection and were not convinced otherwise by the author's rebuttal.

**Justification For Why Not Higher Score:**

All the reviewers gave the paper a confident rejection recommendation.

**Justification For Why Not Lower Score:**

N/A

---

### Decision · Program_Chairs · 2024-01-16

Reject